# Self-Improvement of Non-autoregressive Model via Sequence-Level Distillation

**Yusheng Liao[1], Shuyang Jiang[3], Yiqi Li[1], Yanfeng Wang[1,2], Yu Wang[*1,2]**

[1]Cooperative Medianet Innovation Center, Shanghai Jiao Tong University
[2]Shanghai Artificial Intelligence Laboratory
[3]Fudan University
{liao20160907,17-adamant,wangyanfeng, yuwangsjtu}@sjtu.edu.cn
shuyangjiang23@m.fudan.edu.cn

## Abstract

Although Non-autoregressive Transformer (NAT) models have achieved great success in terms of fast inference speed, this speedup comes with a performance drop due to the inherent *multi-modality* problem of the NAT model. Previous works commonly alleviate this problem by replacing the target side of the raw data with distilled data generated by Autoregressive Transformer (AT) models. However, the multi-modality problem in the distilled data is still significant and thus limits further improvement of the NAT models. In this paper, we propose a method called Sequence-Level Self-Distillation (SLSD), which aims to generate distilled data by the NAT model itself, eliminating the need for additional teacher networks. Furthermore, SLSD can adapt to different NAT models without precise adjustments since the self-distilled data is generated from the same types of NAT models. We conduct extensive experiments on WMT14 EN↔DE and WMT16 EN↔RO and choose five classic NAT models as the backbones to validate the generality and effectiveness of SLSD. The results show that our approach can consistently improve all models on both raw data and distilled data without sacrificing the inference speed.

## 1 Introduction

Non-autoregressive Transformer (NAT) models (Gu et al., 2018) have significantly improved inference speed compared to autoregressive Transformer (AT) models (Vaswani et al., 2017). NAT models make the conditional independence assumption in the output space and generate the entire sequence in parallel. However, this speedup comes at the cost of translation quality due to the *multi-modality* problem (Gu et al., 2018), where the data distribution typically includes numerous potential translations of the same sequence. Unlike AT models, which generate subsequent tokens conditioned

on the previous ones, NAT models lack contextual dependency and are therefore more likely to generate sequences with mixed modalities, leading to a decline in performance.

One way to improve the ability of NAT models to handle complex data is by enhancing their capacity (Ghazvininejad et al., 2020; Qian et al., 2021; Du et al., 2021; Shao et al., 2022). Latent alignment models (Libovický and Helcl, 2018; Saharia et al., 2020) relax the alignment restriction by marginalizing out all monotonic latent alignments using the connectionist temporal classification (CTC) loss (Graves et al., 2006). Recently, Huang et al. (2022c) proposed the Directed Acyclic Transformer (DAT) that can capture multiple translation modalities simultaneously by modeling several decoding paths. Despite the success of these two models, they struggle to handle non-monotonic alignments in machine translation (Shao and Feng, 2022; Ma et al., 2023), which are also common in *multi-modality* problems.

Another common approach to alleviate the *multi-modality* problem is modifying the target sequence (Huang et al., 2022b). One standard method is to simplify raw data with sequence-level knowledge distillation (Kim and Rush, 2016), which replaces the target side of training data with the output of AT models. However, AT distilled data still has two issues. First, the *multi-modality* issue in distilled data still remains (Zhou et al., 2020). Second, Zhou et al. (2020) find that higher-quality distilled data does not necessarily improve the performance of NAT models generally. Weaker NAT models achieve the best performance on distilled data generated by smaller AT, while stronger NAT models prefer the data generated by stronger AT. This phenomenon suggests that, when simplifying data for training a specific NAT model, high quality may not be the first consideration, but rather making the data more suitable for NAT to learn. Considering the aforementioned problems, some

---
[*]Corresponding author

recent works focus on modifying the distillation strategy to generate more adaptive distilled data for NAT models. For instance, Guo et al. (2021) construct the distilled data in five steps via re-ranking and filtering the data using an AT model, and Shao et al. (2022) generate multi-reference distilled data with multiple AT models. However, these works introduce a redundant pipeline and only achieve limited performance improvements.

Therefore, most existing works only consider knowledge distillation as a necessary data processing technique, rather than producing adaptive data for NAT models to learn better. In this paper, we propose a simple yet effective approach to generate distilled data that is more adaptive for NAT models to learn, named Sequence-Level Self-Distillation (SLSD). Inspired by Shao et al. (2022) and Sun and Yang (2020), which suggest that targets with a higher likelihood of the NAT models have fewer problems with *multi-modality*, we search for self-distilled targets in the space of the output distributions produced by the NAT models. We define a score function to select the self-distilled targets to ensure the quality of the self-distilled data. In the SLSD framework, the NAT model itself produces the distilled data, eliminating the need for additional teacher networks. Furthermore, it can adapt to the training processes of different NAT models without requiring precise adjustments.

To validate the effectiveness and generality of the proposed SLSD approach, we conduct extensive experiments on four machine translation benchmarks: WMT14 EN↔DE and WMT16 EN↔RO. We chose five classic NAT methods as the baseline methods, including VNAT (Gu et al., 2018), CMLM (Ghazvininejad et al., 2019), GLAT (Qian et al., 2021), CTC, and DAT. The results show that the SLSD approach can consistently improve all models for different translation directions on both raw and distilled data. Additionally, under the same training strategy, the model fine-tuned on self-distilled data can achieve better performance than the raw data and distilled data generated by AT. The experiments indicate that self-distilled data can mitigate the *multi-modality* problem in raw and distilled data, thereby improving the performance of the NAT models. Our code is released at https://github.com/BlueZeros/SLSD_NAT.

The major contributions of our paper are summarized as follows:

- We propose a simple yet effective method, SLSD, to generate the distilled data by NAT models itself, which can significantly alleviate the *multi-modality* problem in the data and be more adaptive for NAT models to learn.

- We further explore the application of SLSD on various NAT models and find that the proposed framework can be directly applied to raw data without sacrificing inference speed or relying on additional teacher networks.

## 2 Preliminary

In this section, we first briefly describe the task formulation and then introduce three types of NAT models. The machine translation task can be formally defined as a sequence-to-sequence generation problem. Given the target language sequence $\boldsymbol{y}=\{y_1, y_2, ..., y_T\}$ and source language sequence $\boldsymbol{x}=\{x_1, x_2, ..., x_S\}$, the non-autoregressive models assume conditional independence between the output tokens and factorize the output probabilities as $p_{\boldsymbol{\theta}}(\boldsymbol{y}|\boldsymbol{x})=\prod_{t=1}^{T} p(y_t|\boldsymbol{x})$, where $\boldsymbol{\theta}$ represents the parameters of the NAT models.

### 2.1 Vanilla Non-autoregressive Models

Vanilla NAT models are typically trained with cross-entropy loss to maximize the likelihood of the training data:

$$\mathcal{L}_{CE} = -\sum_{t=1}^{T} \log p_{\boldsymbol{\theta}}(y_t|\boldsymbol{x}) \qquad (1)$$

However, the conditional independence assumption makes it difficult for vanilla NAT models to learn directly from raw data. So some works attempt to improve the modeling ability of NAT models by adding contextual information to the inputs:

$$\mathcal{L}_{CE} = -\sum_{t=1}^{T} \log p_{\boldsymbol{\theta}}(y_t|\Omega(\boldsymbol{y}), \boldsymbol{x}) \qquad (2)$$

where $\Omega(\boldsymbol{y})$ is a function to generate the input context. For example, $\Omega(\boldsymbol{y})$ randomly samples words from $\boldsymbol{y}$ and masks these sampled words in the inputs in CMLM and GLAT further adaptively controls the sampling number according to the distance between the output and target sequences.

### 2.2 Connectionist Temporal Classification

Instead of the strict position-to-position calculation in the cross-entropy loss, CTC models a flexible

monotonic alignment between the output sequence and the target sequence. There are two differences between CTC models and NAT models: 1) The input length of the CTC model is typically $\lambda$ times the length of the source sentence. 2) CTC models are allowed to output a "blank" token. With these unique features, the output of the CTC models can align with the target sequence by removing all consecutive repeated tokens and the "blank" tokens. Assume that $\beta(\boldsymbol{y})$ is the set of all possible alignments between the output sequence and target sequence, the training object of the CTC models is calculated by marginalizing the likelihoods of all possible alignments:

$$\mathcal{L}_{CTC} = -\log \sum_{\boldsymbol{b} \in \beta(\boldsymbol{y})} p_{\boldsymbol{\theta}}(\boldsymbol{b}|\boldsymbol{x}) \qquad (3)$$

## 2.3 Directed Acyclic Graph

Previous NAT models hardly handle the multi-modality problem in the raw data. Directed Acyclic Transformers (DAT) attempt to address this issue by stacking a directed acyclic graph (DAG) on the top of the NAT decoder, where the vertices and edges in DAG correspond to hidden states of the decoder and the transitions between the hidden states respectively. The transitions between the connecting vertices constitute multiple possible decoding paths, allowing DAT to capture multiple translation modalities simultaneously. The path probability $p_{\boldsymbol{\theta}}(\boldsymbol{a}|\boldsymbol{x})$ is factorized based on the Markov hypothesis:

$$p_{\boldsymbol{\theta}}(\boldsymbol{a}|\boldsymbol{x}) = \prod_{i=1}^{|\boldsymbol{a}|} p_{\boldsymbol{\theta}}(a_{i+1}|a_i, \boldsymbol{x}) \qquad (4)$$

where $|\boldsymbol{a}|$ is the DAT output length and typically $\lambda$ times the length of the source sequence. Once path $\boldsymbol{a}$ is determined, token $y_i$ can be generated conditioned on the decoder hidden state with index $a_i$. And the DAT can be trained by minimizing the negative log-likelihood loss as below:

$$\mathcal{L}_{DAT} = -\log \sum_{\boldsymbol{a} \in \Gamma} p_{\boldsymbol{\theta}}(\boldsymbol{y}|\boldsymbol{a}, \boldsymbol{x}) p_{\boldsymbol{\theta}}(\boldsymbol{a}|\boldsymbol{x}) \quad (5)$$

where $\Gamma$ is all possible output paths with the same length of target sequence $\boldsymbol{y}$.

## 3 Method

In this section, we first explain the motivation of the proposed SLSD framework. Then we describe the process of generating self-distilled targets in two steps: the sampling of the self-distilled targets and the selection of the self-distilled targets. Finally, we will discuss the training details of the SLSD framework.

### 3.1 Motivation

Previous studies have shown that the data distilled by AT models may not be simple enough for NAT models to learn from, and that high-quality distilled data does not necessarily lead to improved performance of the NAT models in general. This can be attributed to two main reasons: 1) There is a mismatch between the modeling types of NAT models and AT models. The distilled data generated by AT models in an autoregressive manner may not be the most suitable for the learning process of NAT models (Guo et al., 2021). 2) It is challenging to balance the quality and suitability of the distilled data. While higher data quality can prevent information loss, it also indicates that the distilled data is more complex and has a more serious *multi-modality* problem (Zhou et al., 2020). To address these issues, we propose using NAT models as distillation teachers instead of AT models, which can alleviate the mismatch problem. Additionally, our proposed method allows for the control of the quality and suitability of the self-distilled data by adjusting the size of the candidate set, as discussed later in Section 5.1. To prevent confusion, in this paper, we refer to the data generated by the AT models as distilled data and the data generated by our method as self-distilled data.

### 3.2 Sampling process of the SLSD

**Vanilla NAT** Sampling all possible combinations of the whole vocabulary is computationally forbidden as there are a total of $|\mathcal{V}|^T$ samples, where $|\mathcal{V}|$ is the vocabulary size. Instead, we sample $N$ candidates from the output distributions of NAT models to form the candidate set for the selection of the self-distilled targets. A candidate $\boldsymbol{h}$ in the candidates set $\mathcal{H}(x)$ is sampled from the distribution as below:

$$\boldsymbol{h}^{\text{nat}} \sim \prod_{t=1}^{T} p_{\boldsymbol{\theta}}(h_t^{\text{nat}}|\boldsymbol{x}) \qquad (6)$$

**CTC** Unlike vanilla NAT models, the tokens in the outputs of CTC models are not totally conditionally independent, which makes it difficult to calculate the probability of the output sequences. Specifically, the probability of the candidate is the

marginalization of all possible corresponding alignment sequences:

$$h^{\text{ctc}} \sim \sum_{\boldsymbol{b} \in \beta(\boldsymbol{h}^{\text{ctc}})} p_{\boldsymbol{\theta}}(\boldsymbol{b}|\boldsymbol{x}) \quad (7)$$

However, the corresponding alignment set $\beta(\boldsymbol{h^n})$ is exponentially large, making Equation 7 intractable. Alternatively, we sample the alignment sequence $\boldsymbol{b}$ from the output distribution of CTC models to approximate the candidates sampling process:

$$\boldsymbol{b} \sim p_{\boldsymbol{\theta}}(\boldsymbol{b}|\boldsymbol{x}) \quad (8)$$

Then we can get the candidates with the collapsing function $\boldsymbol{h}^{\text{ctc}} = \beta^{-1}(\boldsymbol{b})$ (Saharia et al., 2020). Note that multiple different alignments may correspond to the same candidate during sampling, so the candidate set may contain duplicate samples.

**DAT** Similar to CTC models, it is intractable to marginalize all possible output paths for the candidates. So we factorize the probabilities into the production of the output paths and the output tokens and thus sample the output sequence from the DAT models following a two-step sampling process. The decoding paths are sampled from the transition distributions:

$$\boldsymbol{a} \sim \prod_{i=1}^{|\boldsymbol{a}|} p_{\boldsymbol{\theta}}(a_{i+1}|a_i, \boldsymbol{x}) \quad (9)$$

Then the output tokens are sampled based on the decoding path to get the output sequence $\boldsymbol{f}$:

$$\boldsymbol{f} \sim p_{\boldsymbol{\theta}}(\boldsymbol{f}|\boldsymbol{a}, \boldsymbol{x}) \quad (10)$$

Finally, we can get the candidates with the collapsing function $\boldsymbol{h}^{\text{dag}} = \beta^{-1}(\boldsymbol{f})$.

### 3.3 Selection Process of the SLSD

It is intuitive that samples in the candidate set sampled from the output distribution of the NAT models are the ones that NAT models prefer and are easy to learn. In this section, we mainly focus on selecting the high-quality sample in the candidate set.

We use a score function $\text{score}(\boldsymbol{y}, \boldsymbol{h})$ to measure how similar the candidate $\boldsymbol{h}$ is to the reference $\boldsymbol{y}$. If a sample in the candidate set is close to the reference, we can assume it is high-quality. Our implementation uses $n$-gram overlap as the function to measure the distance between two sequences. We define the set of non-repeating $n$-grams in the

target sequence as $G_n(\boldsymbol{y})$, and the number of times each $n$-gram $\boldsymbol{g} \in G_n(\boldsymbol{y})$ appears in $\boldsymbol{y}$ as $C_{\boldsymbol{g}}(\boldsymbol{y})$. Therefore, the number of $n$-gram matches between the candidate and reference can be defined as:

$$M_n(\boldsymbol{y}, \boldsymbol{h}) = \sum_{\boldsymbol{g} \in G_n(\boldsymbol{y})} \min\left(C_{\boldsymbol{g}}(\boldsymbol{y}), C_{\boldsymbol{g}}(\boldsymbol{h})\right) \quad (11)$$

and the total number of $n$-gram in the reference and the self-distilled data can be defined as:

$$M_n(\boldsymbol{y}) = \sum_{\boldsymbol{g} \in G_n(\boldsymbol{y})} C_{\boldsymbol{g}}(\boldsymbol{y}) \quad (12)$$

$$M_n(\boldsymbol{h}) = \sum_{\boldsymbol{g} \in G_n(\boldsymbol{h})} C_{\boldsymbol{g}}(\boldsymbol{h}) \quad (13)$$

Based on the denotation above, the similarity function is defined as the minimum value of $n$-gram precision and recall of the candidate against the reference

$$\text{sim}(\boldsymbol{y}, \boldsymbol{h}) = \frac{\sum_{n=1}^{N} M_n(\boldsymbol{y}, \boldsymbol{h})}{\sum_{n=1}^{N} \min\left(M_n(\boldsymbol{y}), M_n(\boldsymbol{h})\right)} \quad (14)$$

where $N$ is the maximum size of $n$-gram. Considering candidates sampled from the output distribution of CTC and DAT models have different lengths, we further add a length penalty to constrain the lengths of the candidates:

$$\text{BP}(\boldsymbol{y}, \boldsymbol{h}) = e^{-|1-|\boldsymbol{h}|/|\boldsymbol{y}||} \quad (15)$$

where $|\boldsymbol{h}|$ and $|\boldsymbol{y}|$ are the length of the candidate and reference, respectively. Note that when adopted on the vanilla NAT models, the length penalty is equal to 1. Finally, the score function can be formulated as below:

$$\text{score}(\boldsymbol{y}, \boldsymbol{h}) = \text{BP}(\boldsymbol{y}, \boldsymbol{h}) \cdot \text{sim}(\boldsymbol{y}, \boldsymbol{h}) \quad (16)$$

With the score function to measure the quality of the candidates, we choose the sample with the highest score as the self-distilled targets $\boldsymbol{r}$:

$$\boldsymbol{r} = \underset{\boldsymbol{h} \in \mathcal{H}(\boldsymbol{x})}{\arg\max} \, \text{score}(\boldsymbol{y}, \boldsymbol{h}) \quad (17)$$

### 3.4 Training Schedule

To make sure the candidates sampled from the output distribution of NAT models are meaningful and close enough to the reference, NAT models that have been pretrained on the data are chosen as the initialization of the model in SLSD framework. Then we adopt the self-distilled targets sampling

| Categories | Models | WMT14 | | WMT16 | | Speedup |
| | | EN-DE | DE-EN | EN-RO | RO-EN | |
|---|---|---|---|---|---|---|
| AT | Transformer (Vaswani et al., 2017) | 27.74 | 31.09 | 34.16 | 34.46 | ×1.0 |
| VNAT | VNAT (Gu and Kong, 2021) | 11.40 | 16.47 | 24.52 | 24.79 | ×15.3 |
| | VNAT (Ours) | 11.42 | 15.85 | 24.79 | 24.44 | ×15.3 |
| | SLSD$_{\text{VNAT}}$ | **23.69** | **28.20** | **31.30** | **31.76** | ×15.3 |
| CMLM | CMLM$_1$ (Ghazvininejad et al., 2019) | 10.64 | - | 21.22 | - | ×15.3 |
| | AXE (Ghazvininejad et al., 2020) | 20.40 | 24.90 | 30.47 | 31.42 | ×14.2 |
| | OaXE (Du et al., 2021) | 22.40 | 26.80 | - | - | ×14.2 |
| | *ngram*-OaXE (Du et al., 2022) | 23.60 | 27.90 | - | - | ×15.3 |
| | CMLM$_1$ (Ours) | 11.67 | 15.32 | 21.17 | 22.28 | ×15.3 |
| | SLSD$_{\text{CMLM}}$ | **24.81** | **28.35** | **31.30** | **32.31** | ×15.3 |
| GLAT | GLAT (Qian et al., 2021) | 19.42 | 26.51 | - | - | ×15.3 |
| | LatentGLAT (Bao et al., 2022) | **24.71** | **29.16** | - | - | ×11.3 |
| | GLAT (Ours) | 19.56 | 26.03 | 30.58 | 31.76 | ×15.3 |
| | SLSD$_{\text{GLAT}}$ | 24.66 | 28.77 | **31.21** | **32.27** | ×15.3 |
| CTC | CTC (Libovický and Helcl, 2018) | 18.42 | 23.65 | - | - | ×14.6 |
| | CTC+GLAT (Qian et al., 2021) | 25.02 | 29.14 | - | - | ×14.2 |
| | CTC+DSLP (Huang et al., 2022a) | 24.81 | 28.33 | - | - | ×14.0 |
| | CTC+GLAT (Ours) | 24.73 | 28.79 | 31.03 | 32.06 | ×14.2 |
| | SLSD$_{\text{CTC}}$ | **26.17** | **29.77** | **32.27** | **32.96** | ×14.2 |
| DAT | DAT ($\lambda = 4$) (Huang et al., 2022c) | 26.16 | - | - | - | ×14.2 |
| | DAT ($\lambda = 8$) (Huang et al., 2022c) | 26.57 | 30.68 | - | - | ×13.9 |
| | DAT ($\lambda = 4$) (Ours) | 26.06 | 30.38 | 32.67 | 33.02 | ×14.2 |
| | SLSD$_{\text{DAT}}$ | **26.76** | **31.41** | **33.04** | **33.42** | ×14.2 |

Table 1: The performance of the baseline models and SLSD models on WMT14 EN↔DE and WMT16 EN↔RO raw data. The baseline model with (Ours) indicates our re-implementation. CMLM$_n$ refers to the CMLM model that inferences with $n$ iterations. The best performance of each type of NAT model is bold. SLSD$_{\text{model}}$ represents our method with a different backbone.

and selection pipeline described above to generate the self-distilled data.

For each step in the self-distillation process, we sample from the current output distribution of the model to generate self-distilled targets, which can ensure that the quality of the targets and the performance of the model are synchronously updated. For the decoder input of the NAT models, previous works found that giving some context in the input helps the learning process of NAT models. In contrast, we adopt full masked sequences as the input of the NAT decoders for the targets in the self-distillation process is easy to learn and can reduce the context mismatch between the training and inference process.

## 4 Experiments

### 4.1 Settings

**Datasets** We conduct experiments on both directions of two standard machine transla-

tion benchmarks: WMT14 English↔German (EN↔DE, 4.5M sentence pairs) and WMT16 English↔Romanian (EN↔RO, 0.6M sentence pairs). For WMT14 EN↔DE, we preprocessed the datasets with a joint BPE with 40K merge operations following the pipelines provided in the *fairseq* toolkit (Ott et al., 2019). For WMT16 EN↔RO, we use the pre-processed data provided by Lee et al. (2018). Besides, we choose two additional datasets, Quora[1] and ROCStory (Mostafazadeh et al., 2016), to validate the generalization and effectiveness of the SLSD method. In this paper, we evaluate the performance of models using the BLEU metric (Papineni et al., 2002) for all datasets.

**Hyperparameters of the initialization** Following previous works, we adopt the basic implementation of Transformer-*base* for machine translation tasks. Each model consists of a 6-layer encoder and

---

[1]https://quoradata.quora.com/First-Quora-Dataset-Release-Question-Pairs

| Models | WMT14 | |
| --- | --- | --- |
| | EN-DE | DE-EN |
| Transformer (Vaswani et al., 2017) | 27.61 | 31.48 |
| Vanilla NAT (Gu et al., 2018) | 17.69 | 21.47 |
| CTC (Libovický and Helcl, 2018) | 25.52 | 28.73 |
| AXE (Ghazvininejad et al., 2020) | 23.53 | 27.90 |
| OaXE (Du et al., 2021) | 26.10 | 30.20 |
| Seq-NAT (Shao et al., 2021) | 25.52 | 29.91 |
| CTC+GLAT (Qian et al., 2021) | 26.39 | 29.54 |
| $n$-gram-OaXE (Du et al., 2022) | 26.50 | 30.50 |
| CTC+DSLP (Huang et al., 2022d) | 27.02 | **31.61** |
| DAG ($\lambda$=8) (Huang et al., 2022c) | **27.49** | 31.37 |
| CMLM$_1$ (Ours) | 21.06 | 25.36 |
| SLSD$_{\text{CMLM}}$ | **26.50** | **30.36** |
| GLAT (Ours) | 26.11 | 30.51 |
| SLSD$_{\text{GLAT}}$ | **26.36** | **30.74** |
| CTC+GLAT (Ours) | 26.51 | 30.08 |
| SLSD$_{\text{CTC}}$ | **27.15** | **31.35** |
| DAT ($\lambda$=4) | 26.78 | 31.42 |
| SLSD$_{\text{DAT}}$ | **27.22** | **31.75** |

Table 2: The performance of the baseline models and the SLSD models on WMT14 EN↔DE distilled data.

| Model | Quora | ROCStory | |
| --- | --- | --- | --- |
| | BLEU | BLEU-1 | BLEU-2 |
| VNAT | 21.37 | 29.40 | 4.70 |
| SLSD$_{\text{VNAT}}$ | **24.69** | **35.70** | **8.30** |
| CTC | 25.72 | 38.40 | 7.60 |
| SLSD$_{\text{CTC}}$ | **27.19** | **41.60** | **10.30** |

Table 3: The performance of the baseline models and SLSD on other generation tasks. Note that for the CTC methods we adopt $\lambda = 2$ on Quora and $\lambda = 8$ on ROCStory due to the length ratio of the source sequences and the target sequences.

a 6-layer decoder with 8 attention heads. The hidden dimension and feed-forward layer dimension are 512 and 2048, respectively. For optimization, we use the Adam optimizer (Kingma and Ba, 2015) with $\beta = (0.9, 0.98)$. The weight decay is set to 0.01 and the label smoothing (Szegedy et al., 2016) is set to 0.1. The learning rate warms up for 10k steps to $5e-4$ and decays with an inverse square schedule. For WMT16 En↔Ro, we use a dropout rate of 0.3 and a batch size of 32K tokens, while for WMT14 En↔De, we switch to 0.1 and 64K accordingly. All models are trained with 300k steps on both datasets. For Quora and ROCStory datesets, we train Transformer-*small* with dropout rate of 0.3 for 100k steps. Code implementation is based on *fairseq*.

**Finetuning Hyperparameters for SLSD** We select the last checkpoint of pretrained stage as the initialization of the self-training stage. We finetune models on machine translation dataset and other generation task dataset for 100k steps and 30k steps, respectively. The learning rate is set to $2e-6$ for all models. The batch size of tokens and the dropout rate are set following the corresponding pretrained stage configuration. The size of the candidate set is 100 for DAT models and 40 for the other models. The $n$-gram size of the score function is set to 4.

**Inference** For CMLM and GLAT, we use Length Parallel Decoding (LPD) (Wei et al., 2019) with $l = 5$ length candidates during inference. For CTC models, we directly decode the output sequence. For decoding steps in DAT, we use `lookahead` decoding algorithm. We select the 5 best checkpoints on the validation sets and average them as the evaluation checkpoint.

## 4.2 Main Results

Table 1 shows the BLEU scores for each baseline and our methods on the WMT14 EN↔DE and WMT16 EN↔RO raw datasets. We validated the efficacy of our method on five types of models: VNAT, CMLM, GLAT, CTC, and DAG. The results indicate that our method considerably improves all models on all datasets. Self-distillation can boost CMLM models to exceed previous CMLM-based methods and even achieve higher performance compared to CTC-based models. Specifically, our framework can improve CTC+GLAT models by an average of more than 1 BLEU score, which is a considerable improvement for a strong NAT baseline on raw data. Moreover, the CTC+GLAT model can achieve a BLEU score of 26.17 on WMT14 EN↔DE raw data, which even surpasses DAT ($\lambda$=4) with only $\lambda$=2. Since training DAT ($\lambda$=8) requires significant computing resources and time, we only implement our method on DAT ($\lambda$=4), which still outperforms DAT ($\lambda$=8) by up to 1.03 BLEU score. The effectiveness of our method verifies that the self-training method can significantly mitigate the multi-modality problem in the raw data.

Besides, the improvement of our methods can be seen not only in the raw data but also in the distilled data, which is already simple for NAT models to learn. The performance of our methods and base-

| Models (Steps) | WMT14 | |
| --- | --- | --- |
| | **EN-DE** | **DE-EN** |
| GLAT (300k) | 19.56 | 26.03 |
| *w/* raw data (100k) | 19.45 | 25.62 |
| *w/* distilled data (100k) | 23.92 | **28.81** |
| *w/* self-distilled data (100k) | **24.66** | 28.77 |
| CTC+GLAT (300k) | 24.73 | 28.79 |
| *w/* raw data (100k) | 23.67 | 27.43 |
| *w/* distilled data (100k) | 25.69 | 29.60 |
| *w/* self-distilled data (100k) | **26.17** | **29.84** |

Table 4: The performance of continually training with different data on WMT14 EN↔DE raw data.

line models on WMT14 EN↔DE distilled data are shown in Table 2. All baseline models can benefit from SLSD methods with an average of more than 1 BLEU score gain. The improvements reflect that self-training data generated by the same NAT models is more adaptive for the learning process of various NAT models. Furthermore, we also conduct the experiments on other generation tasks. As the results shown in Table 3, the SLSD can also improve VNAT and CTC methods on Quora and ROCStory dataset, which again validate the generalization and effectiveness of the SLSD method.

### 4.3 Effectiveness of Self-Distilled Targets

In order to eliminate the gain effect brought by more training steps and fairly demonstrate the effectiveness of the proposed SLSD method, we continually train each baseline for another 100k steps. Since the models trained on raw data and distilled data without the help of the GLAT training strategy will result in performance degradation (Qian et al., 2021), we adopt the GLAT ratio $f_{\text{ratio}}$ as the minimum one in the pretrained stage for the self-distillation process. Specifically, we adopt $f_{\text{ratio}}$=0.3 for GLAT and $f_{\text{ratio}}$=0.2 for CTC+GLAT. The results are shown in Table 4. It can be seen that continually training pretrained models on raw data worsen the models' performance, which shows that simply increasing the number of training steps cannot improve the performance of the model. In contrast, distilled data and self-distilled data can simplify the learning process of the NAT models and thus help NAT models achieve higher scores. Specifically, the performance of models finetuned on self-trained data exceeds the models finetuned on distilled data in most language pairs, which again shows that the self-distilled data is more suitable for NAT model

learning.

## 5 Analysis

| $|\mathcal{H}(\boldsymbol{x})|$ | score($\boldsymbol{y}, \boldsymbol{r}$) | NCM$_{\mathcal{X}}(\mathcal{R})$ | **BLEU** |
| --- | --- | --- | --- |
| 0 | - | - | 18.20 |
| 2 | 32.2 | **0.79** | 22.81 |
| 5 | 33.8 | 0.83 | 23.02 |
| 10 | 35.2 | 0.93 | 23.25 |
| 40 | 37.4 | 1.09 | **23.32** |
| 100 | **38.7** | 1.18 | 23.25 |

Table 5: Ablation study on the size of the candidates set during self-distillation process. The experiments are conducted on the WMT14 EN-DE validation set.

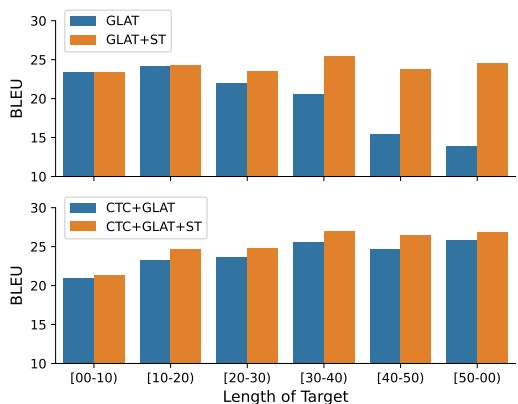

Figure 1: The performance of the generated translations with respect to the lengths of the target sequences. Results are reported on the WMT14 EN-DE test set.

### 5.1 Size of Candidates Set

We conduct the ablation study about the size of the candidates set and show its impact on the self-distillation process. The self-distilled targets set and the self-distilled corpus are denoted as $\mathcal{R}$ and $(\mathcal{X}, \mathcal{R})$, respectively. We measure the complexity of the self-distilled targets set by the Normalized Corpus-level Multi-modality (NCM) for NAT models (Sun and Yang, 2020), which can be calculated as:

$$\text{NCM}_{\mathcal{X}}(\mathcal{R}) = \frac{\mathbb{E}_{(\boldsymbol{x}, \boldsymbol{r}) \sim (\mathcal{X}, \mathcal{R})}[-\log p_{\boldsymbol{\theta}}(\boldsymbol{r}|\boldsymbol{x})]}{\mathbb{E}_{\boldsymbol{r} \sim \mathcal{R}}[|\boldsymbol{r}|]} \quad (18)$$

For data with more complex and serious multimodal problems, NAT models tend to capture more possible translations simultaneously and consequently give rise to reduce probabilities in the models and an increase in NCM.

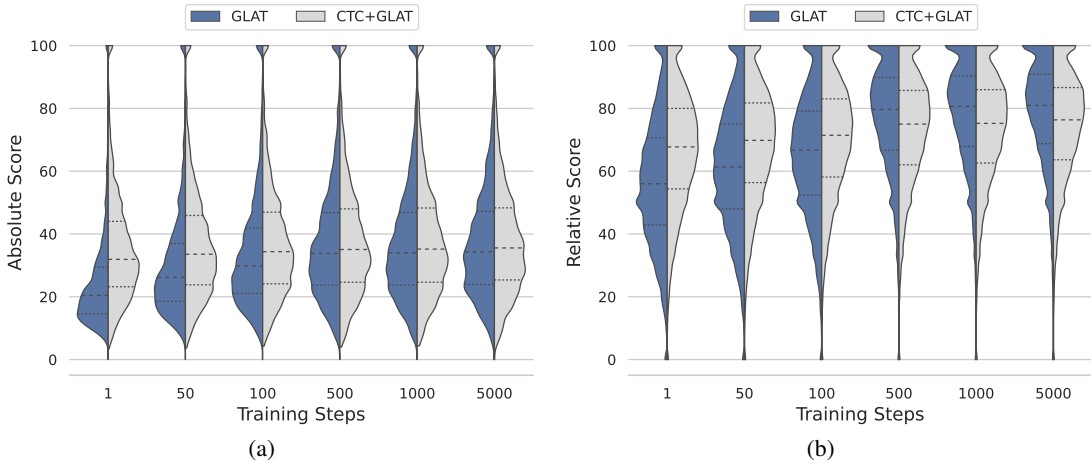

(a)                                                        (b)

Figure 2: The distribution of (a) the Absolute Score and (b) the Relative Score of candidates on two baselines, GLAT and CTC+GLAT, in the first 5k steps of the self-distilled process. The three dashed lines in each violin diagram from top to bottom represent the upper, middle, and lower quartiles, respectively.

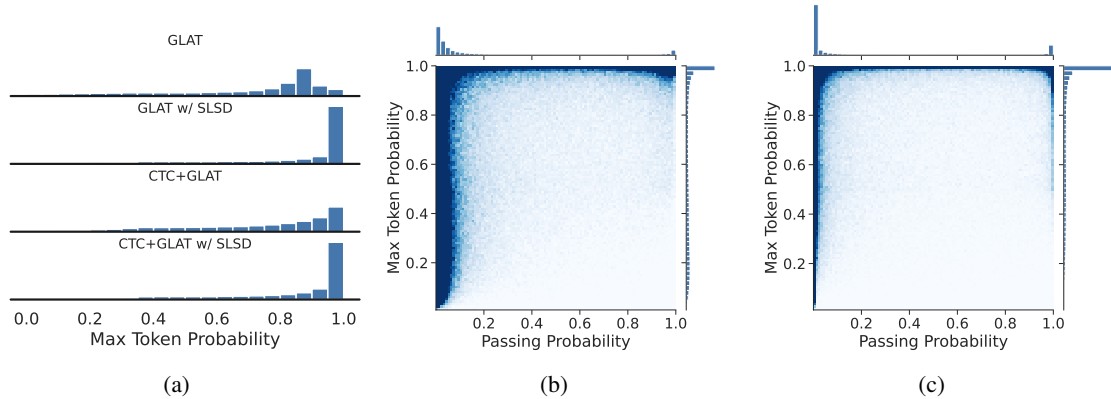

(a)                                        (b)                                        (c)

Figure 3: Output distribution of three GLAT-based NAT models. (a) shows the effect of the SLSD on GLAT and CTC+GLAT models. The difference between the joint distribution of passing probability and max token probabilities can be seen in the output distribution of (b) DAT and (c) SLSD$_{\mathrm{DAT}}$. Darker areas have higher probability values.

The results are shown in Table 5. As the size of the candidate set increases, the score of the self-distilled target and the NCM of the self-distilled targets set are both increased. This shows that a larger candidate set can generate self-distilled targets with higher quality but more complexity. We can achieve a balance between the quality and complexity of the self-distilled data by controlling the size of the candidate set, thereby adapting to the training of the NAR model while ensuring data quality. After tuning on the development set, we can select the best number of candidates set as 40.

## 5.2 Impact on Sequence Length

We also investigate the effectiveness brought by self-distillation on distinct target sequence lengths. To this end, we split the test set into six buckets based on the reference sentence lengths in a range of 10. Figure 1 illustrates the results of adopting SLSD on GLAT and CTC+GLAT backbones in the WMT14 EN-DE test set. It is found that GLAT can only handle short sequences, while SLSD bridges the performance gaps between different lengths and achieve consistent performances across all target lengths. For CTC+GLAT, SLSD can improve its performance in all cases. These results show that SLSD can adapt well to the training of different types of NAT models in different target sentence lengths.

## 5.3 Candidates Score Distribution

We calculate the score distribution of the candidates set in the first 5k steps to better understand the self-distillation process. We use two metrics,

i.e., Absolute Score and Relative Score, to observe the distribution of scores of the candidates set from two orthogonal aspects. The Absolute Score is the maximum score of the candidate sets and reflects the distribution of the whole self-distilled data. The Relative Score is calculated as the ratio of each score to the maximum score in the candidate set, which can measure the distribution of the scores in each candidate set. As shown in Figure 2, both the absolute score and the relative score rapidly increase and stabilize at the beginning of the 500 steps. After the stabilization of the distribution, the absolute score of the GLAT is lower than the CTC+GLAT on average, showing that the self-distilled targets of the CTC+GLAT are better. In contrast, the relative score of the GLAT is higher than CTC+GLAT, which shows that the GLAT is more prone to converge on self-distilled data and CTC has more diverse candidates.

### 5.4 Output Distributions of NAT models

To verify that the SLSD suits the NAT models, we calculate the output distributions of three GLAT-based NAT models and present the result in Figure 3. After training on the self-distilled data, the max token probabilities of GLAT and CTC+GLAT are improved. Besides, for DAT, the passing probabilities are close to 0 or 1, and token probabilities are close to 1. This indicated that SLSD can reduce the *multi-modality* problem in the data, and thus improve the probabilities of the output path in the DAT.

## 6 Conclusion

In this paper, we introduced a simple yet effective method, SLSD, to generate distilled data using NAT models themselves. This approach can greatly reduce the *multi-modality* problem in the data, and consistently improves the performance of four types of NAT models across all datasets. Ablation experiments verify that self-distilled data is better suited for NAT models to learn from, compared to the distilled data generated by AT models.

## Limitations

The proposed SLSD method can produce self-distilled data that is better suited for learning by NAT models than the distilled data generated by AT models, which does not need additional teacher models. However, to obtain relevant and high-quality candidates from the output distribution of NAT models, a well-initialized model is necessary. Moreover, selecting self-distilled targets from the candidate set involves computing the score of each candidate, which requires additional training time and computational cost.

## Acknowledgements

This work is supported by the National Key R&D Program of China (No. 2022ZD0162101), Shanghai Science and Technology Committee (No. 21511101100), STCSM (22DZ2229005), 111 plan (No. BP0719010). We would like to thank all of the anonymous reviewers for their helpful comments.

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
