# OpenReview forum: "Self-Improvement of Non-autoregressive Model via Sequence-Level Distillation"
_EMNLP/2023/Conference — EMNLP 2023 Main_

### Official Review · Reviewer_Gn82 · 2023-08-05

**Soundness:** 4

**Excitement:**

4: Strong: This paper deepens the understanding of some phenomenon or lowers the barriers to an existing research direction.

**Paper Topic And Main Contributions:**

This paper proposes a method called Sequence-Level Self-Distillation (SLSD) to improve non-autoregressive translation models. The key idea is to generate distilled training data using the NAT model itself and then select good ones with an n-gram-based score function. This helps create training data better suited to the NAT model and alleviates the multi-modality problem. The method is evaluated on 4 popular NAT models - CMLM, GLAT, CTC, and DAT on WMT14 En-De and WMT16 En-Ro, and shows consistent and significant improvements over all models on both raw and distilled data.

**Reasons To Accept:**

1. The paper proposes a simple and effective method that works for multiple NAT models without much tuning.
2. The proposed method helps mitigate the multi-modality problem better than distillation with autoregressive models.
3. The paper provided detailed analysis and ablation studies to understand the effect of self-distillation.

**Reasons To Reject:**

1. The proposed method requires a well-initialized NAT model to generate good candidates. For example, it might not work for Vanilla NAT, which is not evaluated in the paper. But this is already acknowledged in the Limitations section.
2. The design of the selection function does not seem very principled (n-gram-based). It is well-known that the n-gram-based metric (e.g., BLEU and ROUGE) has a high variance for sequence-level evaluation.

But overall, this is a nice extension over prior work that shows how self-distillation can further simplify the training data and improve NAT models. The consistent gains across models and datasets are promising.

**Reproducibility:**

3: Could reproduce the results with some difficulty. The settings of parameters are underspecified or subjectively determined; the training/evaluation data are not widely available.

**Reviewer Confidence:**

4: Quite sure. I tried to check the important points carefully. It's unlikely, though conceivable, that I missed something that should affect my ratings.

---

> ### Author Rebuttal · Authors · 2023-08-29
>
> **Q1：The proposed method requires a well-initialized NAT model to generate good candidates. For example, it might not work for Vanilla NAT, which is not evaluated in the paper. But this is already acknowledged in the Limitations section.**
>
> A1：Thank you for your comments. We have conducted the experiments of Vanilla NAT on WMT14 EN-DE raw data, which has similar performance compared with CMLM.
> | **Model** | **WMT14 EN-DE** |
> |:---------:|:---------------:|
> | VNAT      | 10.34           |
> | SLSD_VNAT | **23.69**           |
>
> Besides, our method does not have strict requirements for model initialization. For example, CMLM models only achieve 10.64 BLEU Score on WMT14 ENDE data, which is similar to the performance of Vanilla NAT. However, our method can still improve CMLM to 24.81 BLEU score that even surpasses GLAT. The shortcoming of our model is that it must undergo two-stage training, which causes inconvenience. We will avoid such confusion in the revised paper. Sorry for the confusion and we will modify this statement.
>
> **Q2：The design of the selection function does not seem very principled (n-gram-based). It is well-known that the n-gram-based metric (e.g., BLEU and ROUGE) has a high variance for sequence-level evaluation.**
>
> A2：Thank you for your great comments. We have carefully considered this issue during the developments. We attempted various methods to select the target for the self-distillation, like using AR models or language models. However, due to the fact that our selection function needs to be called at each training step, the selection function based on neural networks can significantly increase the training time. In addition, since the candidate set is generated by the NAR model, the better selection function for self-distillation targets is still limited by the performance of the model itself, which means that a better selection function will not have too much gain on the final performance. Using n-gram based metrics can also be consistent with the testing metrics, reducing the gap between training and testing. Therefore, we adopted the simplest method to select the target of self distillation to strike a balance between performance and efficiency.

---

### Official Review · Reviewer_orC1 · 2023-08-05

**Soundness:** 3

**Excitement:**

3: Ambivalent: It has merits (e.g., it reports state-of-the-art results, the idea is nice), but there are key weaknesses (e.g., it describes incremental work), and it can significantly benefit from another round of revision. However, I won't object to accepting it if my co-reviewers champion it.

**Paper Topic And Main Contributions:**

This paper proposes a method to tackle the issue multi-modality of non-autoregressive transformer models (NAT). From methods of replacing the target side of the raw data with distilled data generated by Autoregressive Transformer (AT) models, this paper proposed generating distilled data from NAT model itself, eliminating external models to generate the data. Results on several translation tasks show the improvement.


**Reasons To Accept:**

1, The method is simple but useful to improve the NAT models, without using external models


**Reasons To Reject:**

1, It is not clear how the proposed method solve the multi-modality issue
2, The methods need to be evaluated in various tasks

**Reproducibility:**

3: Could reproduce the results with some difficulty. The settings of parameters are underspecified or subjectively determined; the training/evaluation data are not widely available.

**Reviewer Confidence:**

4: Quite sure. I tried to check the important points carefully. It's unlikely, though conceivable, that I missed something that should affect my ratings.

---

> ### Author Rebuttal · Authors · 2023-08-29
>
> **Q1：It is not clear how the proposed method solves the multi-modality issue 2**
>
> A1: Thank you for your comments.  As mentioned in Section 3.1 of the paper, the multimodal problem in the NAT model is caused by the NAT model's inability to model the distribution of complex text data. The proposed method mainly solves multi-modality in two aspects:
> * The first is generating the distilled data with NAT models. It is easier for the models to learn because it is generated by the same type of models.
> * The second is is easy to balance data quality and complexity by adjusting the size of candidate sets.
> These two points make our method alleviate the multi-modality problem and can easily adapt to different NAT models.
>
> The experiment results shown in Sections 4.3 and 5.1 indicate that the proposed method mainly improves the NAT models with these two aspects.
>
> **Q2: The methods need to be evaluated in various tasks**
>
> A2：Thank you for your comments. We supplemented the results of the proposed method on two additional tasks, including Quora (Paraphrase Generation) and ROCStory (Story Generation) [1]. Considering the size of the dataset, we adopt transformer-small as the backbone. All the models are training on the date with 100k steps and our method adopts an extra 30k steps self-training process. The batch size is equal to 8k, and the dropout rate is 0.3. The remaining hyperparameters are the same as in other situations.  Specifically, due to the length ratio of the source sequences and target sequences, we adopt λ=2 for Quara and λ=8 for ROCStory, where λ is the upsampling ratio of the CTC. The results are shown below:
> | **Model** | **Quora** | **ROCStory (BLEU1/BLEU2)** |
> |:---------:|:---------:|:--------------------------:|
> | CTC       | 25.72     | 38.40/7.60                 |
> | SLSD_CTC  | **27.19**     | **41.60/10.30**                |
> | VNAT      | 21.37     | 29.40/4.70                 |
> | SLSD_VNAT | **24.69**     | **35.70/8.30**                 |
>
> [1] Huang F, Ke P, Huang M. Directed Acyclic Transformer Pre-training for High-quality Non-autoregressive Text Generation[J]. arXiv preprint arXiv:2304.11791, 2023.

---

### Official Review · Reviewer_nVwZ · 2023-08-12

**Soundness:** 3

**Excitement:**

3: Ambivalent: It has merits (e.g., it reports state-of-the-art results, the idea is nice), but there are key weaknesses (e.g., it describes incremental work), and it can significantly benefit from another round of revision. However, I won't object to accepting it if my co-reviewers champion it.

**Paper Topic And Main Contributions:**

This study solves the problem that high quality distilled data from Autoregressive Transformer Models does not necessarily lead to improved performance of the Non-Autoregressive Transformer Models. The high quality data from AT models is complex and has a multi-modal problem so that it is not suitable for NAT models to fit it easily.

The main contributions of this study are :
1. Use of sequence-level self-distilled data from the NAT models so that it is more suitable for NAT models to fit it.
2. Selection of the higher quality samples in the candidate set from the self-distilled data via a similarity measure with the reference target sample.

**Reasons To Accept:**

It is a clever idea to select good samples from self-distilled data using the reference target, because it is more suitable for fitting NAT models and it is better samples to follow than the self-distilled data itself. This simple method shows good results in several NAT methods.

**Reasons To Reject:**

* The improvements are relatively small for sophisticated NAT methods (GLAT, CTC, DAT). This means that the method in this study is not so effective for the state-of-the-art NAT methods.
* The idea is similar to "Self-Distillation Mixup Training for Non-autoregressive Neural Machine Translation" (https://arxiv.org/abs/2112.11640) in that the both studies use distilled data from NAT models and use reranking and filtering. (But the method in this study is simpler and more effective).

**Reproducibility:**

4: Could mostly reproduce the results, but there may be some variation because of sample variance or minor variations in their interpretation of the protocol or method.

**Reviewer Confidence:**

2: Willing to defend my evaluation, but it is fairly likely that I missed some details, didn't understand some central points, or can't be sure about the novelty of the work.

---

> ### Author Rebuttal · Authors · 2023-08-29
>
> **Q1：The improvements are relatively small for sophisticated NAT methods (GLAT, CTC, DAT). This means that the method in this study is not so effective for the state-of-the-art NAT methods.**
>
> A1: Thank you for your comments. Firstly, We must declare that the proposed method can improve GLAT with 4.9 BLEU scores and CTC with 1.44 BLEU scores on WMT EN-DE data, which is already a significant improvement in machine translation. Besides, considering the proposed method mainly improves the performance of NAT models by alleviating multimodal problems, it is reasonable for some state-of-the-art models that can already handle multimodal problems to have relatively small improvements (but still can improve DAT models with up to 1.03 BLEU score). Our method also enables the model to perform better with fewer memory requirements. For example, CTC (λ=2) with our approach can achieve better performance than DAT (λ=4) and DAT (λ=4) with our method can achieve better performance than DAT (λ=8).
>
> **Q2：The idea is similar to Self-Distillation Mixup Training for Non-autoregressive Neural Machine Translation in that both studies use distilled data from NAT models and use reranking and filtering. But the method in this study is simpler and more effective.**
>
> A2：Thank you for your comments. This paper has been cited in our paper. There are two differences in total between our method and SDMRT (Self-Distillation Mixup Training for Non-autoregressive Neural Machine Translation):
> * SDMRT still uses the distilled data generated from AT models to improve the NAT models. In our method, the distilled data is entirely generated by the NAT model itself, which indicates that NAR models can achieve good performance without relying on the AR model and thus become more efficient and convenient.
> * SDMRT needs to process the training data with a sophisticated framework before the training process, and the data remains unchanged during the training process. In contrast, our method resamples at every step of training to generate new data, which ensures synchronous growth of data quality and model performance.

---

### Meta-Review · Area_Chair_wmWL · 2023-09-17

**Recommendation:** 4

**Metareview:**

The paper proposes a self-distillation method for training non-autoregressive models. The idea is simple, effective and well-evaluated with extensive experiments.

---

### Decision · Program_Chairs · 2023-10-07

**Decision:**

Accept-Main

**Comment:**

The paper proposes a self-distillation method for training non-autoregressive models. The idea is simple, effective and well-evaluated with extensive experiments.